# Biopsychosocial risk factors and knowledge of cervical cancer among young women: A case study from Kenya to inform HPV prevention in Sub-Saharan Africa

Irene Ngune[1], Fatch Kalembo[1], Barbara Loessl[2], Lucy W. Kivuti-Bitok[3]*

**1** Faculty of Health Sciences, School of Nursing Midwifery and Paramedicine, Curtin University, Perth, Western Australia, Australia, **2** College of Science, Health, Engineering and Education (SHEE), Discipline of Nursing, Murdoch University, Murdoch, Western Australia, Australia, **3** School of Nursing Sciences, University of Nairobi, Nairobi, Kenya

* LKIVUTIBITOK@GMAIL.COM, LUKIBITOK@UONBI.AC.KE

## Abstract

### Background

Cervical cancer is the second most common female reproductive cancer after breast cancer with 84% of the cases in developing countries. A high uptake of human papilloma virus (HPV) vaccination and screening, and early diagnosis leads to a reduction of incidence and mortality rates. Yet uptake of screening is low in Sub-Saharan Africa and there is an increasing number of women presenting for treatment with advanced disease. Nine women in their twenties die from cervical cancer in Kenya every day. This paper presents the biopsychosocial risk factors that impact on cervical cancer knowledge among Kenyan women aged 15 to 24 years. The findings will highlight opportunities for early interventions to prevent the worrying prediction of an exponential increase by 50% of cervical cancer incidences in the younger age group by 2034.

### Methods

Data from the 2014 Kenya Demographic and Health Survey (KDHS) was analysed using complex sample logistic regression to assess biopsychosocial risk factors of knowledge of cervical cancer among young women aged 15 to 24 years (n = 5398).

### Findings

Close to one third of the participants were unaware of cervical cancer with no difference between participants aged 15–19 years (n = 2716) and those aged 20–24 years (n = 2691) (OR = 1; CI = 0.69–1.45). Social predisposing factors, such as lack of education; poverty; living further from a health facility; or never having taken a human immunodeficiency virus (HIV) test, were significantly associated with lack of awareness of cervical cancer (p<0.001). Young women who did not know where to obtain condoms had an OR of 2.12 (CI 1.72–2.61) for being unaware of cervical cancer. Psychological risk factors, such as low

publicly available data from The DHS Program
were used for this analysis. The data underlying the
results presented in the study are available by
submitting a Data Access Request through the
following website, stating the objectives of the
study and agreeing to the terms and conditions:
https://microdata.worldbank.org/index.php/catalog/
2544. Further data access information is available
under "Access Policy".

**Funding:** The authors received no specific funding
for this work.

**Competing interests:** The authors have declared
that no competing interests exist.

self-efficacy about seeking medical help, and an inability to refuse unsafe sex with husband or partner, perpetuated the low level of awareness about cervical cancer (p<0.001).

## Conclusions

A considerable proportion of young women in Kenya are unaware of cervical cancer which is associated with a variety of social and psychological factors. We argue that the high prevalence of cervical cancer and poor screening rates will continue to prevail among older women if issues that affect young women's awareness of cervical cancer are not addressed. Given that the Kenyan youth are exposed to HPV due to early sexual encounters and a high prevalence of HIV, targeted interventions are urgently needed to increase the uptake of HPV vaccination and screening.

## Introduction

Cancer is the second leading cause of death worldwide with the types of cancer, incidence and mortality rates, and the burden of disease differing significantly between countries [1]. Cervical cancer is the second most common female reproductive cancer after breast cancer [2], with 84% of the cases being reported in developing economies [2]. In Sub-Saharan Africa, cervical cancer is responsible for the highest number of female deaths, with a mortality rate of 23 cases per 100,000 woman-years compared to 2 cases per 100,000 woman-years in the United States of America (USA) [1, 2]. Additionally, it contributes to the largest cause of potential years of life lost due to young age onset in women between 35 and 50 years [2, 3]. Both the higher incidence and mortality rates result from a lack of or low uptake in screening and preventative measures, late diagnosis, cost and availability of treatments and high human immunodeficiency virus (HIV) infection rates [4].

A report by the the International Agency for Research on Cancer, 2015, page E373, shows that of the cancers that affect women in East Africa, cervical cancer has the highest number of new cancer cases (45.7/100,000), followed by breast cancer (33.5/100,000) and oesophageal cancer (9.8/100,000) [5]. In Kenya, cervical cancer poses a great burden on women's health due to the high incidence [6] and poor prognosis [7]. Incidence rates have continued to rise over the years with an estimated tenfold increase between 1998 and 2011 (414 per 100,000 women in 2011 compared to 48 per 100,000 in 1998) [6]. Data from 2018 shows that cervical cancer contributes 5,250 (12.9%) of new cancer cases and 3,286 (11.84%) of cancer deaths [6]. Incidence rates for women aged 15 to 24 years are predicted to exponentially increase by 50% by 2034 [8]. Currently, nine women in their twenties die of cervical cancer every day [9].

Cervical cancer is mostly caused by a persistent infection with twelve specific carcinogenic types of human papillomavirus (HPV) which is transmitted sexually by males and females shortly after the onset of sexual activity [1, 10]. The two most common high-risk HPV types that cause cervical cancer in women are 16 and 18, which are responsible for approximately 70% of cervical cancers worldwide [11, 12].

Whilst most HPV infections are transitory, in some women they persist, especially if they are HIV positive [1, 10]. Strategies to combat incidence and mortality of cervical cancer include HPV vaccines as primary prevention, and cervical screening to identify early changes as secondary prevention [10]. Despite differing screening and immunisation programs, this has led to a marked decline of cervical cancer and deaths in most developed countries over the

past few decades [1, 10]. Modelling by Hall et al. (2019) [13] shows that incidence rates of invasive cervical cancers in Australia will fall by 42 to 51% over the next 15 years [13]. There are some factors, however, that have led to an increase of cervical cancer rates in younger generations of women in some European and African countries, such as changes in sexual behaviours [14], parity and age at first birth, use of oral contraceptives and tobacco use [15].

Although surveillance programs for cervical cancer for sexually active women and women of reproductive age are in place in Sub-Saharan countries, cervical cancer screening rates have remained relatively low and some of the screening facilities are underutilised [16, 17]. Community awareness of cervical cancer in Kenya may have improved after the introduction of cervical cancer screening programs in 2013 (refer to Box 1 for an overview) and HPV vaccination in 2019 in Kenya [9], but only 14% of women in the reproductive age participate in screening [18] and nearly 50% of women still present with late disease [7]. This leads to twentyfold higher mortality rates in Sub-Saharan women compared to North Africa, Middle East and Europe [1].

---

Box 1. Cervical cancer prevention in Kenya.

Kenyan Cervical Cancer Screening Program [29]

- Target population is women aged 25–49 years (women between 50–65 years can be screened on individual resources)

- Screening interval is 5 years (2 years of HIV positive)

- Types of screening:

  a  Pap smear in women 25–30 years

  b  HPV testing for women >30 years with VIA or VIA/VILI if HPV unavailable (Pap smear if VIA/VILI not possible)

  c  Combined Pap smear and HPV test for HIV positive womenHPV: Cervical swab by health care provider or women herself

VIA and VILI: Visual inspection of the cervix after painting with either with acetic acid (VIA) or Lugol's iodine (VILI). It requires a well-trained health care provider

Pap smear: Scraping cells off the cervix for cytological examination. Requires a laboratory and skilled collectors

Kenyan HPV Vaccination Program [7, 28, 38]

- Commenced November 2019

- Administered to 10 year old girls (target population 9–14 years)

- Two doses 6 months apart

- Quadrivalent (*Gardasil*, covers HPV serotypes 6, 11, 16, 18) and bivalent formulations (*Cervarix*, types 16, 18)

---

Multifactorial biopsychosocial risks lead women to delay screening and treatment [19, 20]. Authors of a Kenyan study reported that poverty, lack of confidence in orthodox medicine, and lack of access to health care were the main reasons for poor uptake of cervical cancer services among women with cervical cancer [19]. The World Cancer Report 2020 [2] outlines further issues, such as a low human development index (HDI) in almost all Sub-Saharan countries, access to anti-cancer drugs and radiation facilities and low physician-to-population ratio.

Understanding all the contextual issues specific to cervical cancer prevention and screening can give insights into various ways policymakers and researchers could address the low uptake and its negative health outcomes. While previous studies on determinants of cervical cancer awareness in Kenya have focused on the provided results for women in the reproductive age-group (15–49 years) [20], and cervical cancer determinants in older women [19, 21] there are limited population based studies that focus on the issues and needs of younger women in the pre-screening age groups (15–24 years). Similarly, primary HPV vaccination in Kenya has targeted adolescents aged 9–14 years [22, 23]; hence the 15-24-year-olds are less catered for by programs.

Young women aged 15–24 years, would be an ideal target group for behavioural interventions to prevent cervical cancer in Kenya because of their developmental stage and the fact that most are sexually active [24]. Given that the health care needs of this age group are different from those of older women, it is important to understand the various biopsychosocial risks young women face, and how they shape their views on cervical cancer screening. The findings of this study will inform the development of cervical cancer preventive strategies that are specific to this group as evidence shows that women are less likely to develop cervical cancer when preventive interventions are implemented before the age of 26 [25]. Our study seeks to untangle and identify factors that contribute to knowledge levels on cervical cancer among this age group. It also builds on the findings of previous studies on cervical cancer in Kenya, and is consistent with the recommended targets of effective cervical cancer prevention such as community mobilisation, outreach, and health education as outlined in chapter 3 of the World Health Organisation (WHO) Comprehensive Cervical Cancer Control Guide [26].

## Biopsychosocial risk factor model and cervical cancer awareness

Our study is guided by Engel's biopsychosocial risk factor model of disease that explains health outcomes as a result of an interaction between biological, social and psychological risk factors [27]. These factors can either predispose, i.e. expose, a person to disease, or perpetuate, i.e. worsen, poor health outcomes by reducing their self-efficacy to deal with poor health [28]. An exploration of the predisposing and perpetuating risk factors for disease is key to understanding a person's awareness of a disease, the level of perceived risk to getting the disease, and consequently their treatment seeking behaviours [29]. The biopsychosocial risk factor model has been instrumental in studies related to determinants of sexual health and risk taking behaviours in young people and their self-efficacy in preventing disease [29, 30].

The biopsychosocial framework provided the fundamental context in which our study examined predisposing and perpetuating risk factors that impact on young women's awareness of cervical cancer which determine the uptake of screening checks and HPV vaccination [31]. In applying the model, we included age, parity, and age at first sexual activity as predisposing biological risk factors. The demographic factors wealth index, religion, place of residence, education level, access to health services and resources, and having taken an HIV test were identified as predisposing social risk factors. Perpetuating psychological factors revolved around personal beliefs about HIV and self-efficacy factors such as the ability to refuse unsafe

sex, and perceptions about the personal risk to HIV transmission. The perpetuating social attributes were 'seeking permission to attend to own's health, 'knowing where to get condoms', and the frequency of using mainstream media (newspaper/radio/Television) (Fig 1).

Assessment of the biological, social and psychological risk factors that predispose or perpetuate low awareness of cervical cancer among women of the pre-screening age group (15–24 year-old) is important as it sheds light on issues that need to be addressed or strengthened to aid policy makers in designing age appropriate and timely intervention programs to prevent, what some authors called, the "next Sub-Saharan African epidemic" [18 p.203].

## Materials and methods

### Sample design

This cross-sectional study used data from the 2014 Kenya Demographic and Health Service (KDHS). These surveys are run in partnership with the USA based DHS Program and the ICF, and the Kenyan Ministry of Health and the National Bureau of Statistics. The KDHS is a nationwide survey with a representative sample of the total population. Samples were drawn from clusters of the 2009 Kenya Population and Housing Census which used a master sampling frame, the Fifth National Sample Survey and Evaluation Programme (NASSEP V). The NASSEP V drew clusters from primary sampling units or enumeration areas (EA) within Kenya's 47 counties which were stratified into urban and rural strata. Demographic and Health Surveys aim to have the best representative sample at national, regional and county levels whilst considering a country's budget and logistics [32]. For the 2014 KDHS, a two-stage cluster sampling procedure was used. At first 1,612 EAs with 995 clusters in rural and 617 in urban areas were selected with equal probability. This number is higher than the usual 300–500 EAs commonly used for a health survey [32]. From these EAs 25 households per cluster were chosen randomly, resulting in a total of 40,300 households [18, 32]. To prevent bias, interviewers visited only the preselected households and were not allowed to replace any household during data collection. Additionally, due to the non-proportional sampling of households from clusters, data has was weighted to be representative.

KDHS ethics approval was granted by the National Commission for Science, Technology and Innovation in Kenya (further details on the ethics process can be obtained from 2014 KDHS report [33]. We were granted permission use the KDHS data to conduct this study by the ICF, a USA based organisation that collaborates with and provides technical assistance to the DHS Program and the Kenya National Bureau of Statistics. The Institutional Review Board (IRB)-approved procedures for DHS public-use datasets do not in any way allow respondents, households, or sample communities to be identified. There are no names of individuals or household addresses in the data files. No identifiable information was accessed by the researchers.

### Measures

The 2014 KDHS contains data from household, women, men and children questionnaires. Long women's questionnaires were administered in half of the households and a short version in the other half [18]. The data used for this study was collected in both versions. Instruments were based on model questionnaires developed by the DHS Program and expanded to meet specific information needs for Kenya. During the development of the questionnaires, consultation meetings were held with various stakeholders to have their input. The same format has been used in previous KDHSs. A total of 14,741 women out of the eligible 15,317 were interviewed, i.e. a response rate of 96%. The age range for the women was 15 to 49 years for the whole sample with 5,392 in the 15 to 24 years bracket. Full details of the questionnaire items can be found in the KHDS report [33].

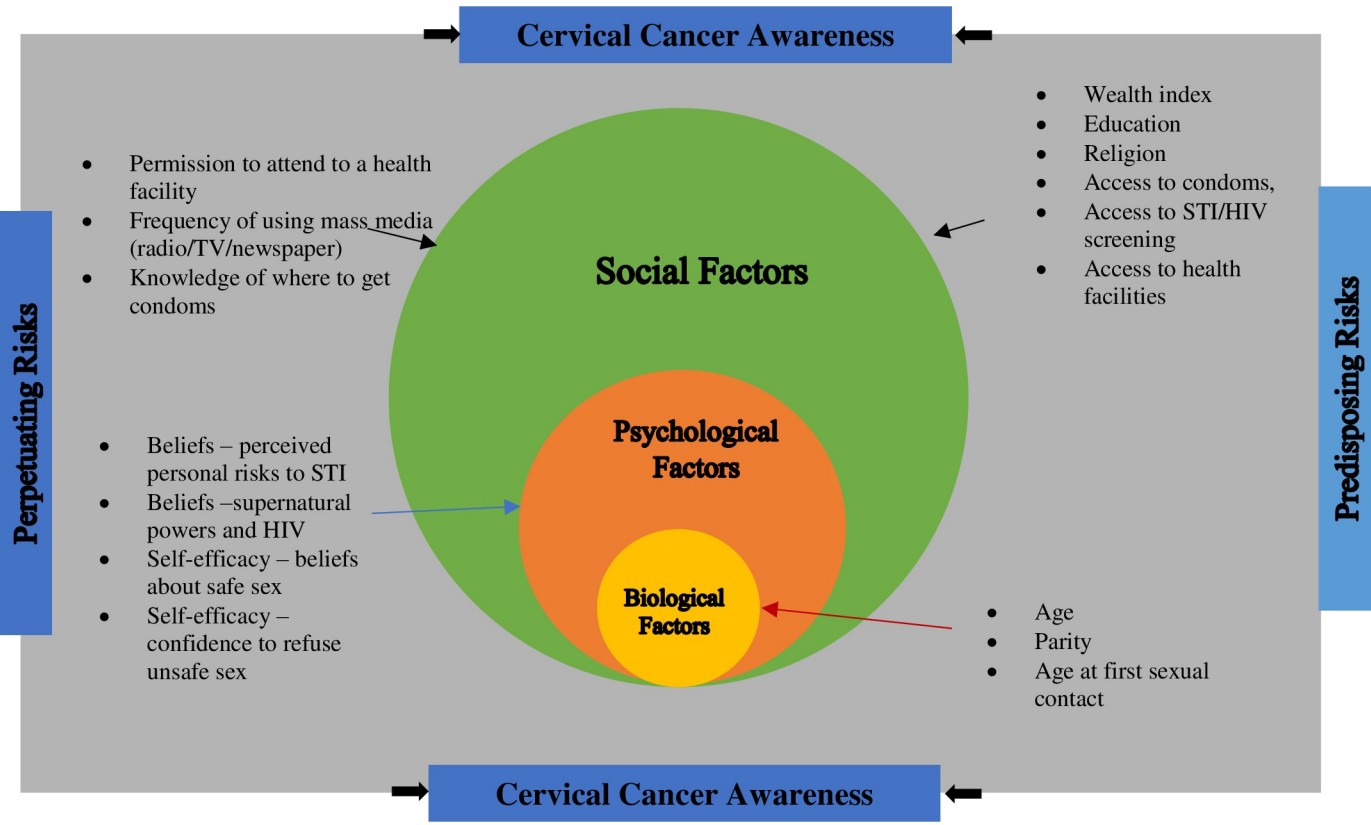

**Fig 1. Biopsychosocial model of risks to cervical cancer awareness.**

## Data items and processing

This study examined the biopsychosocial risk factors for cervical cancer awareness among young women aged 15 to 24 years. This particular age-group was selected because they are in the cervical cancer pre-screening age. Understanding factors that affect their awareness of cervical cancer would be necessary for specific planning of prevention programs for this age-group. Currently, the majority of programs and research in Kenya target adolescents (9–14 years) [34] and women of the entire reproductive age-group [20, 21, 35]. The dependent variable was young women's self-reported awareness of cervical cancer. Within the KDHS questionnaire, the women were asked, 'have you ever heard of cervical cancer?' (No/Yes). The covariates were classified into three levels–biological, social and psychological factors as informed by the conceptual framework. The three levels were grouped into whether the risks were perpetuating or predisposing the young women to limited awareness of cervical cancer. Biological factors were the respondents' age, parity, and age at first sexual activity. The social factors included wealth index, religion, place of residence (urban/rural), education level, distance from a health facility, access to condoms, having taken an HIV test, seeking permission from others to attend a health service, and the frequency of using mass media (newspaper/radio/television). Psychological factors described personal beliefs about HIV ('a healthy looking man can have HIV') and self-efficacy factors, such as the ability to refuse unsafe sex ('a wife can ask a partner or husband to use a condom if they suspect they have a sexually transmitted disease (STI)'), and confidence to seek help ('not wanting to attend a health service alone').

## Data analysis

Since a sub-population of the women's data was used (15–24 years), data was weighted to be representative at the national, regional, and county levels based on women's individual sample weight variable (KDHS variable v005). Both weighted and unweighted data were used to generate frequencies and counts (Table 1).

We used logistic regression with a complex sample analysis function in SPSS to run bivariate and multivariate analyses. The complex analysis function allows multilevel data modelling for cluster and strata sampling units to avoid bias in the standard errors and parameter estimates [36]. Complex analysis was done because the KDHS utilised hierarchical sampling where individual respondents are nested within clusters and regions, which violates the assumption of independence of respondents [36].

For model 1 we entered each individual factor into the model separately to assess their effect on awareness of cervical cancer (unadjusted odds ratios). All exposure variables that were significant at $p \leq 0.25$ in model 1 were entered into the multivariate models 2 or 3 (adjusted odds ratio) to identify factors that were independently associated with unawareness of cervical cancer. Model 2 examined the effect of all predisposing factors classified within the biological (3 variables) and social (10 variables) domains of the biopsychosocial model (Table 2). Model 3 investigated perpetuating social and psychological factors (Table 3). Findings from the three models are presented using adjusted odds ratios (ORs) and 95% confidence intervals. The level of statistical significance was set at $p < 0.05$.

## Results

### Demographic characteristics

Table 1 presents descriptive statistics for both the weighted and unweighted sample. Only weighted results are described. About 30% of young women had not heard about cervical cancer, half of the participants were less than 19 years of age, about two thirds were not in a union, i.e. marriage, and 43.6% had had their first sexual encounter before the age of 19 years. Most were protestant Christians (72%); about 60% had never given birth and lived in rural areas. Most of the women had primary or secondary education (87%), and only 10% were covered by health insurance. The majority had taken an HIV test (72%) but only 39% was able to get condoms. Use of mainstream media was reported as listening to the radio (>80%) and watching television (53%) at least once a week. The majority of women did not have a problem getting medical help due to distance to the health facility (67%), due to money needed for treatment (79%), getting permission to go the health facility (94%) or going alone to the health facility (88%).

### Bivariate and multivariate results

Results from the bivariate and multivariate logistic regression model for the predisposing and perpetuating risk factors are reported in Tables 2 and 3 respectively. Bivariate results informed which variables to include in the multivariate model.

**Predisposing risk factors.** In model 2 we entered both biological and social predisposing variables (factors) with only social predisposing variables showing a stronger association with unawareness of cervical cancer. Women with no education (OR = 24.65, 95% CI [11.66–52.14]), primary education (OR = 3.28, CI [1.73–6.25]), and secondary education (OR = 7.41, CI [3.83–14.34]) were more likely to be unaware of cervical cancer compared to those with higher education. Other social predictors of cervical cancer unawareness were poverty (OR = 1.33, CI [1.08–1.65]), 'cannot or do not know how to get a condom' (OR = 2.12, CI [1.72–2.61]), and 'never been tested for HIV' (OR = 1.44, CI [1.17–1.78]). Biological factors

**Table 1. Descriptive characteristics of the key variables (N = 5392).**

| Variable | | Frequency (%) | |
| --- | --- | --- | --- |
| | Codes | unweighted | weighted |
| **Heard of cervical cancer** | | | |
| No | 0 | 2081 (38.6) | 1752 (32.4) |
| Yes | 1 | 3317 (61.4) | 3655 (67.6) |
| **Respondent's current age in years–M (SD)** | | 19.3 (2.9) | 19.4 (2.9) |
| **Age in 5-year groups** | | | |
| 15–19 years | 1 | 2861 (53) | 2716 (50.2) |
| 20–24 years | 2 | 2531 (47) | 2691 (49.8) |
| **Type of place of residence** | | | |
| Urban | 1 | 1956 (36.2) | 2140 (39.6) |
| Rural | 2 | 3442 (63.8) | 3266 (60.4) |
| **Highest educational level** | | | |
| No education | 0 | 455 (8.4) | 205 (3.8) |
| Primary | 1 | 2353 (47.3) | 2400 (44.4) |
| Secondary | 2 | 2063 (38.2) | 2343 (43.3) |
| Higher | 3 | 327 (6.1) | 458 (8.5) |
| **Religion** | | | |
| Roman Catholic | 1 | 1066 (19.7) | 1058 (19.6) |
| Protestant/other Christian | 2 | 3478 (64.4) | 3899 (72.1) |
| Muslim | 3 | 763 (14.2) | 364 (6.7) |
| No religion/other | 4 | 91 (1.7) | 84 (1.6) |
| **Wealth index** | | | |
| Poorer-poorest | 1 | 2391 (44.3) | 1885 (34.9) |
| Middle | 2 | 1074 (19.9) | 1097 (20.3) |
| Richer-richest | 3 | 1933 (35.8) | 2424 (44.8) |
| **If had a birth** | | | |
| No birth | 0 | 3253 (60.3) | 3276 (60.6) |
| Had at least one birth | 1 | 2145 (39.7) | 2130 (39.4) |
| **Covered by health insurance** | | | |
| No | 0 | 4952 (91.7) | 4842 (89.6) |
| Yes | 1 | 445 (8.2) | 565 (10.4) |
| **Current marital status** | | | |
| Never in union | 0 | 3368 (62.4) | 3433 (63.5) |
| Married/has a partner | 1 | 1825 (33.8) | 1766 (32.7) |
| Widowed/divorced/separated | 2 | 205 (3.8) | 208 (3.8) |
| **Age at first sex** | | | |
| Not had sex | 0 | 2122 (39.2) | 2012 (42.3) |
| 19 and below | 1 | 2414 (44.4) | 2578 (43.6) |
| 20–24 | 2 | 230 (4.2) | 330 (6.1) |
| At first union | 3 | 627 (12.2) | 475 (8.0) |
| **Getting medical help for self: getting money needed for treatment** | | | |
| No problem | 0 | 3411 (63.2) | 3596 (66.5) |
| Big problem | 1 | 1986 (36.8) | 1808 (33.4) |
| **Getting medical help for self: distance to health facility** | | | |
| No problem | 0 | 4102 (76.0) | 4294 (79.4) |
| Big problem | 1 | 1295 (24.0) | 1112 (20.6) |
| **Ever been tested for HIV** | | | |

*(Continued)*

**Table 1.** (Continued)

| Variable | Codes | Frequency (%) | |
|---|---|---|---|
| | | unweighted | weighted |
| No | 0 | 1656 (30.7) | 1483 (27.4) |
| Yes | 1 | 3730 (69.1) | 3915 (72.4) |
| **Can get a condom** | | | |
| No/Don't know | 0 | 3522 (65.2) | 3287 (60.8) |
| Yes | 1 | 1878 (34.8) | 2119 (39.2) |
| **Frequency of reading newspaper or magazine** | | | |
| Not at all | 0 | 3231 (59.9) | 3054 (56.5) |
| < Once a week | 1 | 1150 (21.3) | 1219 (22.5) |
| At least once a week | 2 | 1015 (18.8) | 1132 (20.9) |
| **Frequency of listening to radio** | | | |
| Not at all | 0 | 1138 (21.1) | 862 (15.9) |
| < Once a week | 1 | 777 (14.4) | 737 (13.6) |
| At least once a week | 2 | 3482 (64.5) | 3808 (70.4) |
| **Frequency of watching television** | | | |
| Not at all | 0 | 2951 (54.7) | 2539 (47.0) |
| < Once a week | 1 | 803 (14.9) | 762 (14.1) |
| At least once a week | 2 | 1644 (30.5) | 2105 (38.9) |
| **Getting medical help for self: getting permission to go** | | | |
| No problem | 0 | 4974 (92.1) | 5048 (93.7) |
| Big problem | 1 | 424 (7.9) | 358 (6.6) |
| **Getting medical help for self: not wanting to go alone** | | | |
| No problem | 0 | 4662 (86.4) | 4778 (88.4) |
| Big problem | 1 | 732 (13.6) | 624 (11.5) |
| **Reduce risk of getting HIV: have 1 sex partner only, who has no other partners** | | | |
| No/Don't know | 0 | 636 (11.8) | 515 (9.4) |
| Yes | 1 | 4762 (88.2) | 4868 (90.6) |
| **A healthy-looking person can have HIV** | | | |
| No | 0 | 758 (14.1) | 672 (9.0) |
| Yes | 1 | 4470 (83.6) | 4868 (90.0) |
| Don't know | 3 | 123 (2.3) | 104 (1.0) |
| **Wife justified asking husband to use condom if he has a sexually transmitted disease (STI)** | | | |
| No/Don't know | 0 | 705 (13.1) | 489 (9.1) |
| Yes | 1 | 4414 (81.8) | 4702 (87.0) |
| **Can get HIV by witchcraft or supernatural means** | | | |
| No | 0 | 4918 (91.1) | 5071 (93.8) |
| Yes/Don't know | 1 | 480 (8.9) | 335 (6.2) |
| **Respondent can ask partner to use a condom** | | | |
| No/Don't know/Depends | 0 | 4109 (76.1) | 4030 (74.5) |
| Yes | 1 | 1280 (23.9) | 1377 (25.5) |

such as age group, parity or age at first sex did not have an effect on awareness of cervical cancer.

**Perpetuating risk factors.** Table 3 shows the effect of perpetuating factors on cervical cancer awareness. After controlling for perpetuating social and psychological factors in model 3, almost all of the social and psychological factors had an impact on the level of awareness of cervical cancer among young women. Social factors that worsened awareness of cervical

**Table 2. Adjusted and unadjusted odds ratios of predisposing factors of unawareness of cervical cancer among young women aged 15–24 years.**

| Variable | | |
|---|---|---|
| | Unadjusted OR (95% CI) | Adjusted OR (95% CI) |
| **Predisposing biological factors** | | |
| **Age in 5-year groups (Ref: 20–24)** | | |
| 15–19 years | 2.15 (1.86–2.32) | 1 (0.69–1.45) |
| **If had a birth (Ref: No birth)** | | |
| Had at least one birth | 1.33 (1.13–1.56) | 1 (0.69–1.45) |
| **Age at first sex (Ref: At first union)** | | |
| Not had sex | 1.06 (0.80–1.39) | 0.88 (0.59–1.32) |
| 19 and below | 0.51 (0.39–0.67) | 0.76 (0.54–1.08) |
| 20–24 | 0.15 (0.08–2.72) | 0.55 (0.28–1.06) |
| **Predisposing social factors** | | |
| **Type of place of residence (Ref: Urban)** | | |
| Rural | 2.14 (1.79–2.56) | 1.14 (0.93–1.40) |
| **Highest educational level (Ref: Higher)** | | |
| No education | 77.22 (38.33–155.59) | 24.65 (11.66–52.14) |
| Primary | 15.65 (8.45–29.01) | 7.41 (3.83–14.34) |
| Secondary | 5.95 (3.22–10.98) | 3.28 (1.73–6.25) |
| **Religion (Ref: No religion/other)** | | |
| Roman Catholic | 0.47 (0.28–0.79) | 0.76 (0.43–1.32) |
| Protestant/other Christian | 0.47 (0.28–0.78) | 0.77 (0.46–1.34) |
| Muslim | 1.51 (0.87–2.62) | 1.43 (0.79–2.58) |
| **Wealth index (Ref: Richer-Richest)** | | |
| Poorer-poorest | 2.87 (2.39–3.44) | 1.33 (1.08–1.65) |
| Middle | 1.54 (1.31–2.04) | 1.14 (0.90–1.44) |
| **Covered by health insurance (Ref: Yes)** | | |
| No | 2.52 (1.76–3.61) | 1.30 (0.90–1.87) |
| **Current marital status (Ref: Widowed/divorced/separated)** | | |
| Never in union | 1.27 (0.84–1.93) | 1.32 (0.78–2.23) |
| Married/has a partner | 0.96 (0.62–1.49) | 1.06 (0.64–1.75) |
| **Getting medical help for self: getting money needed for treatment (Ref: No problem)** | | |
| Big problem | 1.46 (1.25–1.71) | 0.99 (0.83–1.19) |
| **Getting medical help for self: distance to health facility (Ref: No problem)** | | |
| Big problem | 1.94 (1.65–2.29) | 1.50 (1.11–2.03) |
| **Can get a condom (Ref: Yes)** | | |
| No/Don't know | 3.36 (2.82–4.01) | 2.12 (1.72–2.61) |
| **Ever been tested for HIV (Ref: Yes)** | | |
| No | 2.66 (2.27–3.13) | 1.44 (1.17–1.78) |

Factors and covariates used in the computation are fixed for all variables in the table: Age in 5-year groups = 20–24; If had a birth = No birth; Age at first sex = At first union; Type of place of residence = Urban; Highest educational level = Higher; Religion = No religion/other; Wealth index = Richer-Richest; Covered by health insurance = Yes; Current marital status = Widowed/divorced/separated; Getting medical help for self: getting money needed for treatment = No problem; Getting medical help for self: distance to health facility = No problem; Can get a condom = Yes; Ever been tested for HIV = Yes.

OR = Odds Ratio; (Ref.) = Reference Category; CI = confidence intervals.

**Table 3. Adjusted and unadjusted odds ratios of perpetuating factors on awareness of cervical cancer among young women aged 15–24 years.**

| Variable | Heard of cervical cancer (Ref: Yes) | |
|---|---|---|
| | unadjusted OR (95% CI) | Adjusted OR (95% CI) |
| Perpetuating Social factors | | |
| Getting medical help for self: getting permission to go (Ref: No problem) | | |
| Big problem | 1.96 (1.50–2.57) | 1.32 (0.97–1.79) |
| Frequency of reading newspaper or magazine (Ref: At least once a week) | | |
| Not at all | 2.72 (2.19–3.39) | 2.03 (1.60–2.59) |
| Less than once a week | 1.53 (1.18–1.99) | 1.38 (1.04–1.83) |
| Frequency of listening to the radio (Ref: At least once a week) | | |
| Not at all | 2.69 (2.21–3.28) | 1.55 (1.23–1.90) |
| Less than once a week | 1.83 (1.49–2.25) | 1.44 (1.15–1.79) |
| Frequency of watching television (Ref: At least once a week) | | |
| Not at all | 3.04 (2.54–3.64) | 2.04 (1.68–2.48) |
| Less than once a week | 1.63 (1.25–2.11) | 1.33 (0.99–1.79) |
| A healthy-looking person can have HIV (Ref: Yes) | | |
| No | 2.42 (1.96–2.99) | 1.74 (1.40–2.18) |
| Don't know | 6.85 (3.95–11.86) | 2.61 (1.52–4.47) |
| Perpetuating Psychological Factors | | |
| Getting medical help for self: not wanting to go alone (Ref: No problem) | | |
| Big problem | 1.78 (1.44–2.20) | 1.30 (1.01–1.69) |
| Reduce risk of getting HIV: have 1 sex partner only, who has no other partners (Ref: Yes) | | |
| No/Don't know | 2.44 (1.97–3.04) | 1.51(1.18–1.94) |
| Wife justified asking husband to use condom if he has STI (Ref: Yes) | | |
| No/Don't know | 4.25 (3.47–5.20) | 2.68 (2.14–3.35) |
| Can get HIV by witchcraft or supernatural means (Ref: No) | | |
| Yes | 2.55 (1.92–3.38) | 1.49 (1.09–2.03) |
| Respondent can ask partner to use a condom (Ref: Yes) | | |
| No/Don't know/depends | 1.64 (1.36–1.97) | 1.48 (1.22–1.80) |

Factors and covariates used in the computation are fixed for all the variables in the table: Getting medical help for self: getting permission to go = No problem; Frequency of reading newspaper or magazine = At least once a week; Frequency of listening to radio = At least once a week; Frequency of watching television = At least once a week; A healthy looking person can have HIV = Yes; Getting medical help for self: not wanting to go alone = No problem; Wife justified asking husband to use condom if he has STI = Yes; Can get HIV by witchcraft or supernatural means = No; Respondent can ask partner to use a condom = Yes.

OR = Odds Ratio; (Ref.) = Reference Categories;; CI = confidence intervals

cancer were a reduced frequency or not using mass media. Women who never read a newspaper (OR = 1.55, CI [1.23–1.90]) or less than once a week (OR = 1.44, CI [1.15–1.79]) were almost one and half times more likely to be unaware of cervical cancer compared to women who read the paper at least once a week. Similarly, those who did not listen to the radio or watch television, were significantly more likely to be unaware of cervical cancer (p ≤.001).

Psychological factors such as confidence to seek medical help alone, and beliefs and practices related to the personal risk of getting HIV were significantly associated with unawareness of cervical cancer. Young women who found it a big problem to seek medical help alone were

1.3 times more likely to be unaware of cervical cancer (OR = 1.30, CI [1.01–1.69]) compared to those with no problem. Similarly, those who felt a wife is not justified to ask a husband to use a condom if he has an STI or reported they had no confidence to ask their partner to use a condom were significantly more likely to be unaware of cervical cancer compared to those who had the confidence to refuse unsafe sex (p ≤.001). Women who were unaware that a healthy-looking man could have HIV were at least 1.7 times more likely to be unaware of cervical cancer compared to those who were aware of such a risk (OR = 1.74, CI [1.40–2.18]). Perceived personal reduced risk to HIV when their partner has other partners (OR = 1.51, CI [1.18–1.94]), and the belief that HIV can be acquired through supernatural means (OR = 1.49, CI [1.09–2.03]) significantly predicted unawareness of cervical cancer.

## Discussion

A significant proportion (32.4%) of young women in our study had never heard of cervical cancer. This is higher than those reported by authors of two previous Kenyan studies which were (20%) [13] and 24% [14] respectively. This difference is likely to due to differences in the age of the participants. The lack of cervical cancer knowledge reported in our study is similar to a Tanzanian study (30.9%) [34] and lower than the results from a Nigerian study (42.7%) [4]. The Tanzanian data was based on a 2011–12 survey, whereas the Nigerian sample was of high school students from 13 to 25 years, surveyed in 2018. There is ample evidence linking the lack of awareness of cervical cancer among young women in Sub-Saharan Africa to low uptake of cervical cancer prevention services, such as HPV vaccination and cancer screening [20, 37, 38], which leads to presentations of advanced cervical cancer and late treatment [11, 23]. In fact, even if there was an awareness of cervical cancer as the KDHS data indicates (76.2% of women 15–49 years), over 80% had never been screened [20]. Our study cohort was below the age of eligibility for screening as the Kenyan program is for women 25 to 49 years (refer to Box 1). We hypothesised that predisposing and perpetuating biological, psychological and social risks factors would predict the level of cervical cancer awareness among the young women in our study sample.

### Predisposing factors

The predisposing social factors, such as lack of education, poverty, no access to condoms, or never having taken an HIV test (27.4%) significantly increased the odds of having low levels of awareness about cervical cancer. These findings were similar to those by Kangmennaang et al. [20], who used the KDHS data but looked at the whole female cohort. While age and a higher parity increased cervical cancer knowledge in their study [20], biological factors such as age, parity and age at first sexual contact had no significant impact on the young women in our study. About 35% of the participants had a wealth index of 'poor to poorest'. According to a report by the World Bank, over one third of the Kenyan population live below the international poverty line of US$1.90 per day [39].

Two main factors stand out–the education level and the ability to access condoms. Over 90% of the participants achieved a secondary level or less with about half completing only primary education. Almost two thirds (60.8%) did not know where to get condoms. The latter, together with the fact that about 44% of women have their first sexual encounter below the age of 19, is quite concerning.

### Perpetuating factors

Perpetuating social and psychological risk factors were varied and have also been previously reported in Kenya and Nigeria studies [4, 17, 20]. Poor evaluations of personal risk to HIV and

low self-efficacy about seeking medical help increased the odds for low awareness. However, just like in other studies in Kenya [19, 20, 35], health literacy determined by access to mass media, such as radio, television and newspapers, has a significant impact on knowledge levels of cervical cancer. The lower the engagement and access, the higher the odds for low awareness. The major challenges faced by women are limited access to media (ownership of radio, television, etc.) and poor literacy related to lower economic status [40].

An interesting finding, which is different from the results of previous studies done in Kenya was the fact that the majority of young women (87%) felt that they were justified to ask their partners with an STI to use a condom, but only 25% thought they actually could demand it. The low self-efficacy in women with regards to safe sex practices predicted higher levels of unawareness of cervical cancer. Behavioural issues related to self-efficacy, such as the inability to refuse unsafe sex, is a significant determinant of cervical cancer awareness for this age-group. Self-efficacy with regards to cervical cancer prevention among Kenyan women has not been extensively explored due to a lack of population-based studies. A qualitative study by Ngugi et al. [21] on factors affecting uptake of cervical cancer screening in Kenya reported self-efficacy in relation to attendance of screening among older women only [21]. To our knowledge, this is the first study in Kenya to identify a significant association between self-efficacy and awareness of cervical cancer.

Some younger women were also reluctant to attend health care facilities on their own and had issues with distance, availability and affordability. Gender inequality plays a major role in power dynamics within a household and ultimately affects how women access screening and other prevention programs for cervical cancer [20].

## Prevention of cervical cancer

Based on our findings of various predisposing and perpetuating factors within a biopsychoso-cial framework, there are a range of strategies that could help with reducing the current and future burden of cervical cancer in Kenya. The most efficient prevention of cervical cancer is the HPV vaccination, followed by HPV screening and cytology [13]. HPV vaccination commenced in Kenya in 2019 with the initial aim to target 800,000 ten-year-old girls [9]. Kivuti-Bitok et al. (2014) assessed the impact of vaccination and screening on cervical cancer rates in Kenya [41]. They found that a secondary vaccination, i.e. targeting women up to 44 years of age who missed the primary vaccination for girls who are not yet sexually active (9–12 years), was the most important intervention, accounting for a more than 50% reduction in incidence rates [41]. This would necessitate education campaigns to make all women aware, a roll-out strategy for those secondary immunisations, and the financial ability to do so. With the introduction of the"100% transition policy" by the Kenyan government for all primary school students to attend high school, prevention programs targeting 15 to 24-year-olds could easily be rolled out through high schools and colleges [40].

Additionally, screening needs to be available for those who have not been vaccinated to allow early diagnosis and treatment. HPV testing is the gold standard for screening but is not always possible to conduct due to financial and logistical reasons [1, 41]. Other challenges that lead to low uptake screening services include fear of being diagnosed with the disease, long distance to the health facility, and shame of undergoing a vaginal examination [26]. Thus, service providers, policymakers and researchers need to consider these issues when planning cancer preventive interventions. Kenya's screening program reflects this by offering alternatives to HPV-based screening (refer to Box 1) [42]. An additional consideration is that an HPV sampling can be done by the woman herself, which might increase the uptake of screening [1]. This might particularly be the case for women with low self-efficacy and gender equity.

While our study targeted the 15 to 24 year-old young women, the other two Kenyan studies targeted women of the whole reproductive age group (15–49 years) [13, 14]. Future studies in Kenya should assess the levels of knowledge of cervical cancer among women aged between 25 to 49 years, as this is the recommended age group for cervical cancer screening programs.

Comprehensive community based cervical cancer awareness campaigns are needed in order to reach out to young women of lower socioeconomic status. Some studies suggested empowering and equipping local health care providers in health centres and dispensaries [43], and school teachers [44] to pass on the message. The authors argued that the majority of girls and women obtained cervical cancer information from health care providers during other related health visits, and from teachers in the local schools [43, 44]. This has been very successful in HIV education and prevention as almost three-quarters of the young women had been tested for HIV, and only a small percentage had low awareness of their risk for infection. Perhaps HPV and HIV campaigns should be linked, starting at an early age.

Additionally, the role of mobile phone messaging in delivering health promotion messages to young women should be explored. Although a report by Wesolowski, et al. in 2012 on mobile phone usage in Kenya showed that individuals who did not use a phone at all were primarily female (81%) and had no education (40%), the study also indicated that there was some level of mobile phone ownership in every level of income bracket with at least 20% in the lowest, i.e. individuals with incomes less than a 1,000 Kenyan shillings (about $10 per month) [45]. It is also plausible that with lower costs and easier availability, the amount of mobile and smartphones will have increased substantially since then.

Self-efficacy is an indication of one's ability and confidence to exert control over their motivation, behaviour, and social environment [46]. Young women who were unable to refuse unsafe sex with their partner or husband, or to seek medical help without the help of others had poor knowledge of cervical cancer. Empowerment of women through education, business and employment opportunities can lead to women participating in decision making on issues that affect their health [47]. Additionally, policies that seek to empower women financially, and those that offer monetary subsidies or free treatment to young women of lower socioeconomic status are warranted. Addressing gender and culturally related barriers to access health services and obtain health-related items, such as condoms, are also key factors in reducing the level of cervical cancer unawareness. Considering that older women, especially those with several births, are already more aware of cervical cancer and screening this could also mean that mothers should play a more significant role in educating their daughters and accompanying them to health care facilities.

Finally, cervical cancer prevention campaigns targeting men are needed, given that culturally they are entrusted with the decision making on issues that affect women's health [48].

## Limitations

We assessed the effect of biological factors of age, parity and age at first sexual contact but did not investigate other clinically relevant factors such as HIV status that are associated with increased prevalence of cervical cancer [49]. Our study did, however, use the variable of 'ever completing an HIV test' to assess awareness of cervical cancer. A population-based data of HIV status of young women in this sample was not available from the KDHS. Furthermore, our study was not able to establish a causal relationship between biopsychosocial risk factors and unawareness of cervical cancer among young women due to the cross-sectional nature of the data.

## Conclusion

The findings of our study underscore the importance of government, non-governmental organisations, health institutions, schools and community leaders to work together to identify

feasible educational programs that target women at risk of being unaware of cervical cancer. Increasing cervical cancer knowledge among young women is a critical step towards improving the uptake of cervical cancer prevention services. Qualitative studies are also warranted to better understand factors that influence awareness of cervical cancer and to identify culturally appropriate interventions that can increase the uptake of preventative measures.

## Supporting information

**S1 File. ICF authority letter.**
(PDF)

**S2 File. Investigators authority letter.**
(PDF)

## Author Contributions

**Conceptualization:** Irene Ngune, Fatch Kalembo, Barbara Loessl, Lucy W. Kivuti-Bitok.

**Data curation:** Irene Ngune, Fatch Kalembo, Barbara Loessl, Lucy W. Kivuti-Bitok.

**Formal analysis:** Irene Ngune, Fatch Kalembo, Barbara Loessl, Lucy W. Kivuti-Bitok.

**Investigation:** Irene Ngune, Fatch Kalembo, Barbara Loessl, Lucy W. Kivuti-Bitok.

**Methodology:** Irene Ngune, Fatch Kalembo, Barbara Loessl, Lucy W. Kivuti-Bitok.

**Resources:** Irene Ngune, Fatch Kalembo, Barbara Loessl, Lucy W. Kivuti-Bitok.

**Validation:** Irene Ngune, Fatch Kalembo, Barbara Loessl, Lucy W. Kivuti-Bitok.

**Visualization:** Irene Ngune, Fatch Kalembo, Barbara Loessl, Lucy W. Kivuti-Bitok.

**Writing – original draft:** Irene Ngune, Fatch Kalembo, Barbara Loessl, Lucy W. Kivuti-Bitok.

**Writing – review & editing:** Irene Ngune, Fatch Kalembo, Barbara Loessl, Lucy W. Kivuti-Bitok.

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
