## [Decision Letter · Decision Letter 0]

30 Jun 2020

PONE-D-20-14074

Biopsychosocial risk factors and knowledge of cervical cancer among young women: A case study from Kenya to inform HPV prevention in Sub-Saharan Africa

PLOS ONE

Dear Dr. KIVUTI-BITOK,

Thank you for submitting your manuscript to PLOS ONE. After careful consideration, we feel that it has merit but does not fully meet PLOS ONE’s publication criteria as it currently stands. Therefore, we invite you to submit a revised version of the manuscript that addresses the points raised during the review process.

The authors should check if they can provide data on smaller age brackets. A core issue are the conclusions (publication criterion 4), please review according to comments made by reviewer 2.

We look forward to receiving your revised manuscript.

Kind regards,

Hajo Zeeb

Academic Editor

PLOS ONE

Journal Requirements:

2. Thank you for providing clarification about ethics approval and for uploading the ICF Authority letter. The letter states that "The IRB-approved procedures for DHS public-use datasets do not in any way allow respondents, households, or sample communities to be identified. There are no names of individuals or household addresses in the data files." Please add this statement to your ethics statement and the text of your methods section, and indicate whether there was any other identifying information that was accessed by the researchers.

Reviewers' comments:

Reviewer's Responses to Questions

**Comments to the Author**

1. Is the manuscript technically sound, and do the data support the conclusions?

Reviewer #1: Yes

Reviewer #2: Partly

2. Has the statistical analysis been performed appropriately and rigorously? 

Reviewer #1: Yes

Reviewer #2: Yes

3. Have the authors made all data underlying the findings in their manuscript fully available?

Reviewer #1: Yes

Reviewer #2: Yes

4. Is the manuscript presented in an intelligible fashion and written in standard English?

Reviewer #1: Yes

Reviewer #2: Yes

5. Review Comments to the Author

Reviewer #1: The authors have presented the manuscript in a manner that is technically sound and it makes meaning when one reads it. Conclusions in the manuscript is adequately backed by the data. Data collected by DHS is very reliable and the authors used appropriate analytical tool to run the analysis.

Reviewer #2: My review is uploaded.

Summary: The results of the evaluation of the subgroup of 15 – 24 year old adolescent girls wand young women within the KDHS survey are not new and do not bring new dimensions which need to be addressed in HPV vaccination and cervical cancer prevention programs. It is therefore not clear how this helps in further refining the national cervical cancer program in Kenya. It might be useful to do analysis at the level of the 47 counties where the survey was performed to identify local difference in knowledge and communities which need special attention.

6. PLOS authors have the option to publish the peer review history of their article (what does this mean?). If published, this will include your full peer review and any attached files.

Reviewer #1: **Yes: **Salome Amissah-Essel

Reviewer #2: **Yes: **Andreas Dr Ullrich

---

## [Author Response · Author response to Decision Letter 0]

15 Jul 2020

The specific reviewers comments are as attached in the rejoinder

---

## [Editor Report · Decision Letter 1]

24 Jul 2020

PONE-D-20-14074R1

Biopsychosocial risk factors and knowledge of cervical cancer among young women: A case study from Kenya to inform HPV prevention in Sub-Saharan Africa

PLOS ONE

Dear Dr. KIVUTI-BITOK,

Thank you for submitting your manuscript to PLOS ONE. After careful consideration, we feel that it has merit but does not fully meet PLOS ONE’s publication criteria as it currently stands. Therefore, we invite you to submit a revised version of the manuscript that addresses the points raised during the review process.

The comments have been addressed, but a few issues remain.

Stating that only adjusted OR are shown is not entirely correct as your tables contain unadjusted OR.

Tables: there is no need for p-values, CI are sufficient and correct as you are presenting estimates. Please remove the stars and stick to CI only.

For Table 2 and 3 please indicate in the legend what the adjustment variables are. The tables need to be able to stand alone, with complete information.

Discussion - please reconsider outlining future studies at the beginning of the discussion, this should be reserved for the final section, and consider all further research needs in combination.

Redo the spell check and check all brackets, e.g. L10 in the discsussion

We look forward to receiving your revised manuscript.

Kind regards,

Hajo Zeeb

Academic Editor

PLOS ONE

---

## [Author Response · Author response to Decision Letter 1]

27 Jul 2020

The comments have been included as attached in the rebuttal letter.

---

## [Editor Report · Decision Letter 2]

3 Aug 2020

Biopsychosocial risk factors and knowledge of cervical cancer among young women: A case study from Kenya to inform HPV prevention in Sub-Saharan Africa

PONE-D-20-14074R2

Dear Dr. KIVUTI-BITOK,

We’re pleased to inform you that your manuscript has been judged scientifically suitable for publication and will be formally accepted for publication once it meets all outstanding technical requirements.

Kind regards,

Hajo Zeeb

Academic Editor

PLOS ONE
---

## [Editor Report · Acceptance letter]

10 Aug 2020

PONE-D-20-14074R2 

Biopsychosocial risk factors and knowledge of cervical cancer among young women: A case study from Kenya to inform HPV prevention in Sub-Saharan Africa 

Dear Dr. KIVUTI-BITOK:

I'm pleased to inform you that your manuscript has been deemed suitable for publication in PLOS ONE. Congratulations! Your manuscript is now with our production department. 

Kind regards, 

on behalf of

Prof. Hajo Zeeb 

Academic Editor

PLOS ONE